

# SORAS, A ground-based 110 GHz microwave radiometer for measuring the stratospheric ozone vertical profile in Seoul

Soohyun Ka[1], Jung Jin Oh[1,2]

[1]Research Institute of Global Environment, Sookmyung Women's University, Seoul, 04310, Korea
[2]Department of Chemistry, Sookmyung Women's University, Seoul, 04310, Korea

**Correspondence:** Jung Jin Oh (jjinoh@sookmyung.ac.kr)

**Abstract.** A ground-based 110 GHz radiometer was designed to measure the stratospheric ozone vertical profile by observing the 110.836 GHz ozone emission spectrum and the instrument has been operational at Sookmyung Women's University (37.54 °N, 126.97 °E) in Seoul, Korea. In this paper, we detail the instrumental design, calibration procedures,

correction methods, and the retrieved ozone vertical profile. The instrument is a heterodyne total power radiometer. It down-converts the observed 110.836 GHz ozone frequency to 0.609 GHz, with a frequency resolution of 61 kHz and a bandwidth of 800 MHz. The spectral intensity is digitized using a fast Fourier transform spectrometer. For hot-cold calibration, we use microwave absorbers at room temperature and liquid nitrogen as calibration targets. Tropospheric opacity is corrected using the continuous tipping curve calibration. The measured opacities were compared with simulated values from the Korea Local

Analysis and Prediction System (KLAPS) data. Additionally, since 2016, the stratospheric ozone profiles over Seoul have been demonstrated for the vertical range of 100 hPa – 0.3 hPa (16 km–70 km) with validation performed by comparing them to the ozone profiles from the MLS on AURA satellite.

## 1 Introduction

Ozone is a trace gas in the atmosphere, with the highest concentrations found in the lower to middle stratosphere. Commonly

referred to as the ozone layer, it is a key factor in global warming and is crucial for absorbing significant amounts of solar UV-B radiation. Ozone is mainly formed by photochemistry in the upper stratosphere (Prather, 1981) and is transported to the rest of the atmosphere. As the ozone in the lower and middle stratosphere has a relatively long lifetime, its distribution is primarily influenced by atmospheric dynamics. The discovery of the ozone hole over the South pole (Farman et al., 1985) was a major global issue, prompting the 1987 Montreal Protocol to limit halogen gas emissions. Consequently, the overall

abundance of ozone-depleting substances (ODSs) in the atmosphere has decreased since the late 1990s, and total ozone in the Antarctic is showing signs of recovery (World Meteorological Organization (WMO), 2022). Likewise, recovery of ozone in the upper stratosphere outside of polar regions has been noted, but trends in the lower and middle stratosphere remain unclear. However, some studies suggest continued ozone depletion in the lower and middle stratosphere over northern mid-latitude. (Steinbrecht et al., 2017; Ball et al., 2018; Bernet et al., 2019; Szeląg et al., 2020). Since the trends in ozone vertical



distibution above specific latitude regions have differed from those of the total ozone column, it is essential to monitor the vertical ozone profile up to the upper stratosphere, and even the mesosphere. This monitoring is crucial for understanding stratospheric dynamics and enhancing future climate change predictions.

The ground-based microwave radiometer in this study is a totally passive instrument that measures the vertical profile in the stratosphere, even up to the lower mesosphere by detecting rotational emissions from trace gases with permanent electric

or magnetic dipole moments such as ozone (Hocke et al., 2007; Fernandez et al., 2015b), water vapor ($H_2O$) (Deuber et al., 2004; Straub et al., 2010; De Wachter et al., 2011; Gomez et al., 2012), chlorine monoxide (ClO) (Solomon et al., 2000; Connor et al., 2013; Nedoluha et al., 2020), nitric oxide (NO) (Newnham et al., 2011), carbon monoxide (CO) (Straub et al., 2013), and so on. The zonal wind profile in the middle atmosphere (Rüfenacht et al., 2012) and the tropospheric temperature and humidity profiles (Massaro et al., 2015) are observed by the ground-based microwave radiometer.

The stratospheric ozone profiles measured by the ground-based microwave radiometer have a high temporal resolution from continuous monitoring. It enables studies on the transport or dynamics of the middle atmosphere by tracking ozone distribution. The diurnal cycles of stratospheric ozone relative to local solar time have been studied with an hourly resolution (Parrish et al., 2014; Studer et al., 2014; Maillard Barras et al., 2020). The polar vortex displacement towards mid-latitudes was studied through short-term fluctuations in ozone measurements taken at 30-minute intervals (Moreira et al., 2018).

Moreover, there are studies for a long-term natural variability in the stratospheric ozone caused by the annual and semi-annual oscillations (AO and SAO), the quasi-biennial oscillation (QBO), El Niño-Southern Oscillation (ENSO), and solar activity cycle (Moreira et al., 2016).

In this paper, we introduce the ground-based 110.836 GHz microwave radiometer named SORAS (Stratospheric Ozone RAdiometer in Seoul), designed to measure the vertical distribution of stratospheric ozone. This instrument detects the

spontaneous radiation emitted during the $6_{15} - 6_{06}$ rotational transition. It has been developed and operated at the Research Institute of Global Environment (RIGE) of Sookmyung Women's University (SMWU, 37.54 °N, 126.97 °E) in Seoul, Korea. Notably, since 2015, RIGE at SMWU has been designated as a commissioned observatory for climate change by the Korea Meteorological Administration (KMA). This designation underscores the critical role of SORAS in monitoring the stratosphere, contributing to national climate change observation.

As the first microwave radiometer developed in Korea for stratospheric monitoring, we have implemented multiple hardware modifications and conducted numerous tests to determine the most effective calibration and data correction methods. When SORAS was first developed in 2008, it was configured as a double-sideband system, which differs from its current configuration. In addition, there were issues with the artifacts on the spectrum and suboptimal data correction. Information on the equipment in its initial state and the data from that period are presented in Ka and Oh (2009, 2012, 2014)

papers.

In the following sections, we describe the stable version of the radiometer post-2016. Section 2 provides instrumental descriptions. Section 3 presents the spectroscopy, calibration, and corrections for SORAS measurement. Section 4 shows the



retrieved ozone profile from SORAS measurements and the validation with AURA Microwave Limb Sounder (MLS) ozone data. Section 5 summarizes the conclusions of this study.

## 2 Instrumentation

SORAS is a 110 GHz single-sideband heterodyne receiver. It operates at room temperature in an indoor laboratory (Fig. 1). The indoor environment enables the instrument to operate continuously, regardless of weather conditions. The radiometer consists of a quasioptical part with mirrors and an antenna, a frontend with microwave components, and a backend with a spectrometer.

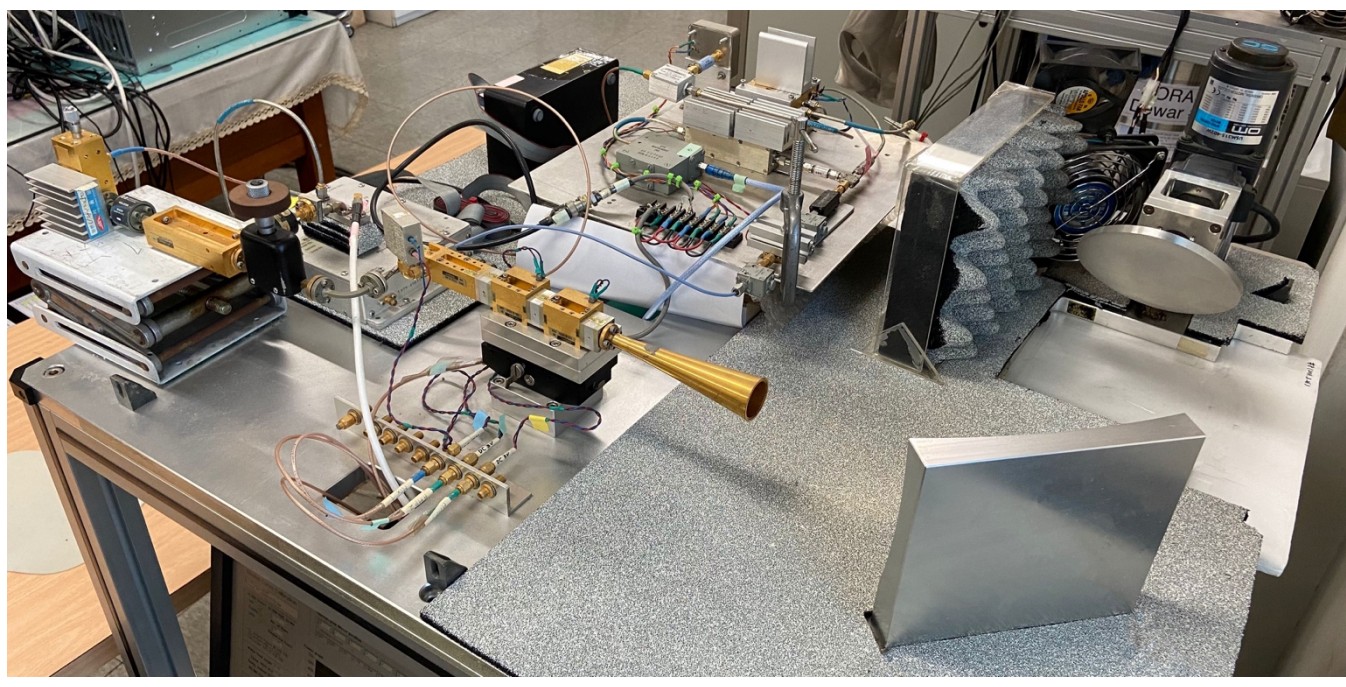

**Figure 1.** SORAS is in operation at the laboratory of Sookmyung Women's University, Seoul. This figure displays the quasioptical system and the frontend; the backend and the control PC are located under the front-end platform.

The details of SORAS are described in Fig. 2. The quasioptical system comprises a rotating plane mirror of an oval shape, an ellipsoidal mirror, and a corrugated horn antenna. The rotating plane mirror controls the selection of the measurement target with a step angle set to 0.18 degrees. This mirror wobbles continuously back and forth during measurements to suppress standing waves between optical components, which are generated by the phase overlap of propagating forward and backward waves. The ellipsoidal mirror, positioned at 45 degrees off-axis, is designed to focus the incident radiation on the corrugated horn antenna, which has a beam waist of 7 mm at 110 GHz. The corrugated horn





antenna was manufactured by Millitech Inc., and its full width at half maximum (FWHM) of 8.3 degrees is given by the antenna pattern measurement for 110 GHz (Fig. 3).

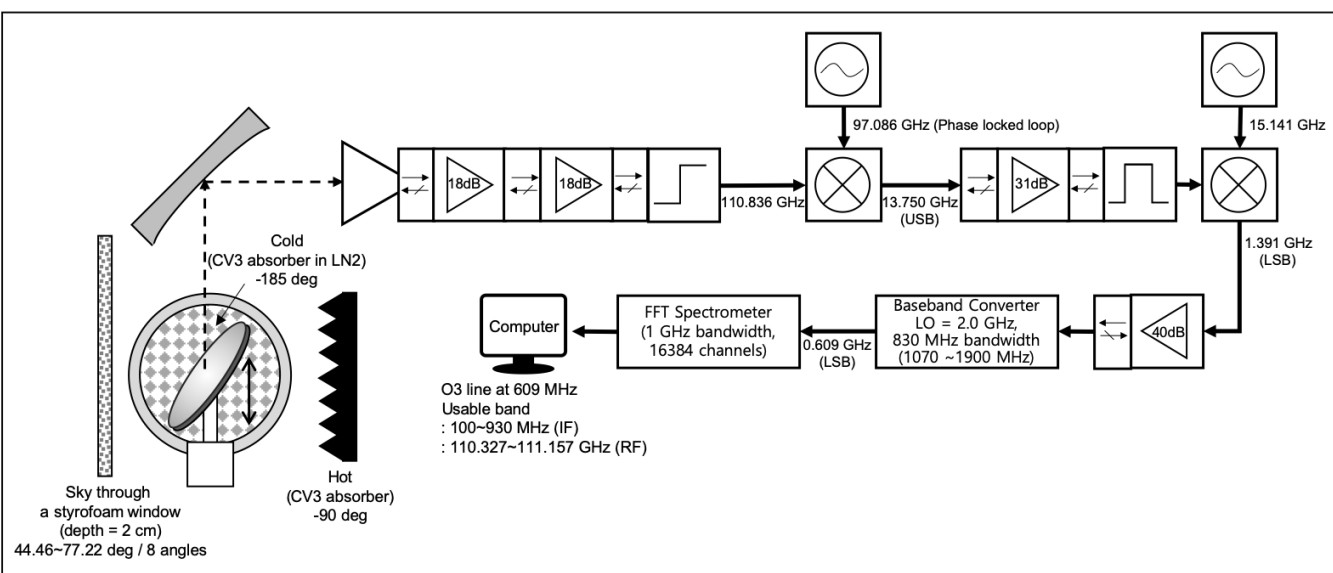

**Figure 2.** SORAS block diagram illustrating the quasioptical parts with calibration targets, microwave components of the frontend, and the backend with FFT spectrometer.

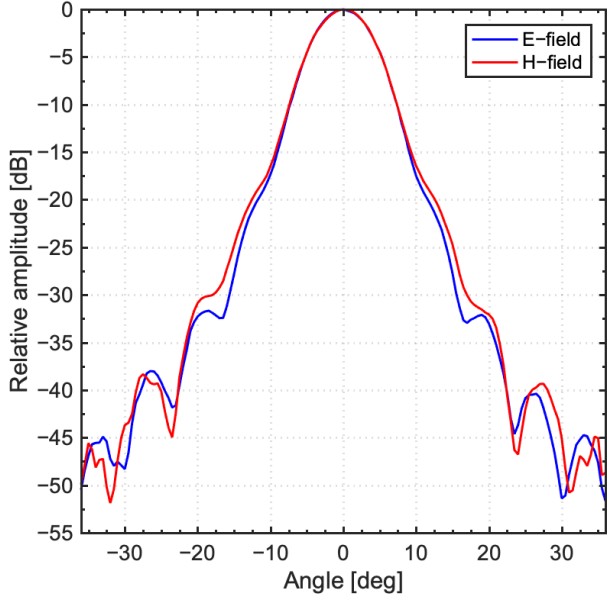

**Figure 3.** SORAS far-field antenna pattern at 110 GHz for the E-plane and H-plane.



The radiation passing through the horn antenna is processed by the front-end. The major purpose of the front-end is to amplify the signal intensity with low noise and to convert to a lower frequency range that can be analyzed with a spectrometer. For the SORAS front-end, the incident signal is amplified twice by two identical low-noise amplifiers of 18 dB. The lower frequency than 100 GHz is cut off by a high pass filter. The frequency of 110.836 GHz is down-converted to the

95 intermediate frequency (IF) of 13.750 GHz by a gunn oscillator. The phase-locked loop (PLL) system is used to verify the 97.086 GHz frequency generated from the gunn oscillator; the PLL uses a harmonic mixer with a 97.076 GHz signal generated from 8.0896 GHz and a 10 MHz reference signal. The IF signal is amplified, filtered, and down-converted again to 1.391 GHz. At the baseband converter, the final frequency of 0.609 GHz is established in 830 MHz bandwidth.

The 0.609 GHz signal is analyzed by a digital fast Fourier transform (FFT) spectrometer of Acqiris AC240 (Benz et al.,

2005) with a full bandwidth of 1 GHz and a frequency resolution of 61 kHz. However, the baseband converter has an asymmetrical 830 MHz bandwidth, and the actual bandwidth is determined here. As a result, considering the bandwidth of the baseband converter, the available frequency range of the received signal is limited from 110.327 GHz to 111.157 GHz. Based on the observed data, the spectrum was available beyond 111.157 GHz, so we used the spectrum from 110.427 GHz to 111.227 GHz in this study.

The additional temperature sensors connected to SORAS provide the ambient temperature and the temperature of a calibration target, which are mentioned in Section 3.2. The meteorological parameters such as temperature, pressure, relative humidity, precipitation, and so on are measured by the Vaisala WXT510 weather transmitter installed on the roof of the building. The system operation and data storage are controlled by LabVIEW software.

## 3   Measurement

### 110   3.1   Spectroscopy

The radiation frequency of atmospheric ozone is determined by the rotational transitions. However, its spectrum becomes broadened during propagation from the emitted frequency due to the collisions with other gases. The random motion of the ozone molecules also contributes to this broadening. These broadening effects are known as pressure broadening and Doppler broadening, respectively. To measure the ozone profile, we focus on the pressure broadening, which is dominant in

the stratosphere and troposphere. To simulate the ozone spectrum, we must understand the absorption coefficient $\alpha$ at the frequency $\nu$:

$$\alpha_{O3} = \sum_{i=1}^{n} \alpha_i(\nu) \tag{1}$$

where $\alpha_i$ is the absorption coefficient of i-th atmospheric molecule. The $\alpha_i$ is consists of the number density $N_i$, spectral intensity $S_j$, and line shape factor $F_j$ at j-th spectral line for the i-th molecule at pressure $p$ and temperature $T$.

$$120 \quad \alpha_i(\nu) = N_i \sum_{j=1}^{m} S_j(T) F_j(\nu, p, T) \tag{2}$$



Here, $N_i$ can be calculated from the partial pressure and temperature of the gases. $S_j$ and $F_j$ are well described in many spectroscopic texts (Gordy and Cook, 1984; Buehler et al., 2005). Given the atmospheric profiles (pressure, temperature, and humidity) and the spectroscopic parameters from the JPL (Jet Propulsion Laboratory) or the HITRAN catalog, we can simulate the atmospheric ozone spectrum. The ozone absorption coefficient is well-defined by spectroscopy, and we can calculate its value based on the theory. However, the rotational spectroscopy for water vapor, liquid water, and oxygen is not as well understood, spanning hundreds of gigahertz. This broad spectrum is referred to as the continuum, and the absorption coefficient has been suggested empirically by the atmospheric propagation model, such as the MPM (Liebe and Layton, 1987; Liebe, 1989; Liebe et al., 1993) or Rosenkranz (Rosenkranz, 1998). As oxygen and water are concentrated in the troposphere, the continuum is often regarded as the tropospheric spectrum.

In the Rayleigh-Jeans approximation, the atmospheric spectrum is described by the radiative transfer equation with the absorption coefficient:

$$T_b = T_{bg} \cdot e^{-\tau} + \int_0^{Z_{top}} T \cdot \alpha \cdot e^{-\tau} dz \tag{3}$$

where $T_b$ is the brightness temperature measured at the ground, $T_{bg}$ is the microwave cosmic background temperature of 2.7 K, and $T$ is the temperature profile of the atmosphere. The opacity $\tau$ is the integral of absorption coefficients along the zenith path $z$:

$$\tau = \int_0^z \alpha(s) ds \tag{4}$$

The atmospheric spectrum of Eq. (3) can be expressed according to the atmospheric layers by defining the weighted mean tropospheric temperature $T_{trop}$ and the brightness temperature of the ozone $T_{b,O3}$ in the middle atmosphere. As the tropospheric ozone is present in very small amounts, its contribution to $T_b$ can be negligible.

$$T_b = \left(T_{bg} + a_{mid} T_{b,O3}\right) e^{-\tau_{tr} a_{tr}} + T_{trop} \left(1 - e^{-\tau_{tr} a_{tr}}\right) \tag{5}$$

where $a_{mid}$ and $a_{tr}$ are air masses in the tropospheric and middle atmospheric, respectively, and $\tau_{tr}$ is the tropospheric zenith opacity. It should be noted that in Eq. (5), $T_{b,O3}$ represents the brightness temperature of ozone as observed at zenith, whereas $T_b$ denotes the brightness temperature obtained at the measurement angle. Air mass depends on the zenith angle of the measurement and Earth's curvature affects the measurement, especially at high zenith angles. In this study, the air masses are calculated according to Eq. (6) and Eq. (7) with $R$ being the Earth's radius (6378 km), $h$ the tropopause height (16 km), and $H$ the height of the middle atmosphere (84 km) (De Wachter et al., 2011).

$$a_{tr} = \frac{\sqrt{(R+h)^2 - R^2 \sin^2(\theta)} - R\cos(\theta)}{h} \tag{6}$$

$$a_{mid} = \frac{\sqrt{(R+h+H)^2 - R^2 \sin^2(\theta)} - \sqrt{(R+h)^2 - R^2 \sin^2(\theta)}}{H} \tag{7}$$

The $T_{trop}$ is weighted mean temperature of the troposphere, assumed to be a homogeneous isothermal single layer. It can be derived from the absorption coefficients by the continuum models with defined atmospheric profiles (Ingold et al., 1998).

$$T_{trop} = \frac{\int_0^h T \cdot \alpha \cdot e^{-\tau} dz}{\int_0^h \alpha \cdot e^{-\tau} dz} \tag{8}$$





If the frequency for the absorption coefficient is far from the ozone transition line, $T_{b,O3}$ in Eq. (5) can be disregarded. Consequently, $T_b$ represents the tropospheric continuum.

The calculated atmospheric spectra in the zenith direction are shown in Fig. 4. The continuum was calculated based on the MPM93 model (Liebe et al., 1993) with the monthly averaged atmospheric profiles (pressure, temperature, and humidity) obtained from a mixture of radiosonde data at Osan, approximately 50 km away from the SORAS location, and the AURA MLS satellite data. The ozone profiles in the middle atmosphere were taken from the MLS data. Ozone transition lines appear as spikes in the spectra, while the continuum looks like a baseline. The continuum at 110 GHz ranges from approximately 50 K to 195 K due to Korea's climate, which features hot, humid summers and cold, dry winters. The

intensity of the ozone spectrum is relatively lower in July (summer) than in January (winter) owing to varying tropospheric opacity.

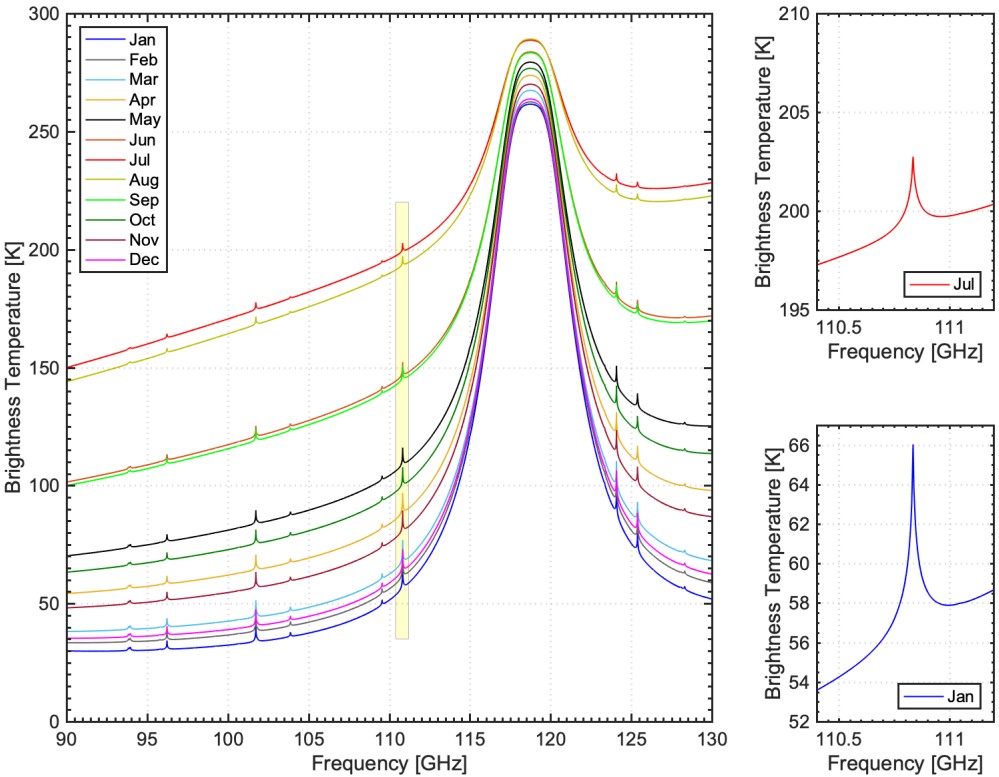

**Figure 4.** The atmospheric spectrum derived from monthly averaged profiles near SORAS location, with the 110.836 GHz ozone transition line to be measured in this study highlighted.



## 3.2 Calibration

In this study, the microwave radiometer is designed to measure the atmospheric spectrum at given frequencies. To convert the output from the FFT spectrometer to brightness temperature, at least two reference signals emitted from loads of different temperatures, ideally blackbodies, are required. This method is known as hot-cold calibration. Using this calibration, the

170 atmospheric brightness temperature $T_b$ is calculated from the measurement by the following equation:

$$T_{b,meas}(\theta) = \frac{T_{hot} - T_{cold}}{V_{hot} - V_{cold}} (V_{atm}(\theta) - V_{cold}) + T_{cold} \tag{9}$$

where $T_{hot}$ and $T_{cold}$ are the temperature of calibration targets, $V_{hot}, V_{cold}$ are the corresponding voltages from the FFT spectrometer, and $V_{atm}$ is the voltage that measures the atmosphere at zenith angle $\theta$. An Eccosorb® CV3 microwave absorber in ambient temperature is used as a hot target, and the identical absorber soaked in liquid nitrogen serves as a cold

target. The receiver noise temperature has been calculated using the y-factor method (Eq. (10), (11)), and it is currently 1540 K for the SORAS system.

$$y_{fac} = \frac{V_{hot}}{V_{cold}} = \frac{T_{hot} + T_{rec}}{T_{cold} + T_{rec}} \tag{10}$$

$$T_{rec} = \frac{T_{hot} - y T_{cold}}{y - 1} \tag{11}$$

### 3.2.1 Temperature correction

As the SORAS system has been operated indoors, it has observed atmospheric signals through a common styrofoam window with a thickness of 2 cm. The transmittance ($t$) of the styrofoam was measured experimentally to estimate the signal loss through the window as $t = 0.997$. With the signal loss factor, the atmospheric brightness temperature is corrected from $T_{b,meas}$ by the following equation:

$$T_b = \frac{T_{b,meas} - (1 - t) T_{air}}{t} \tag{12}$$

where $T_{air}$ is the air temperature measured by the weather transmitter installed on the roof of the building.

The temperature of a cold calibration target is assumed to be the boiling point ($T_{LN2,BP}$) of liquid nitrogen contained in a dewar. It is calculated using the Clausius-Clapeyron equation, considering the air pressure $p$ (Deuber et al., 2004):

$$T_{LN2,BP} = \left[ \frac{1}{T_{0,LN2}} - \left( \frac{R}{L} \right) \ln \left( \frac{p}{p_0} \right) \right]^{-1}$$

$T_{0,LN2}$ (77.3 K) is the boiling temperature of liquid nitrogen at the standard pressure $p_0$ (1013 hPa). $R$ (8.3144 Jmol[-1]K[-1]) is

190 the universal gas constant and $L$ (5660 Jmol[-1]) is the latent heat of evaporation. The air pressure $p$ is obtained from the weather transmitter data.

Considering the reflectivity ($\gamma$) at the surface of liquid nitrogen, the effective boiling temperature is given as:

$$T_{LN2} = T_{LN2,BP}(1 - \gamma) + T_{amb}\gamma \tag{13}$$

where $T_{amb}$ is ambient temperature, and $\gamma$ is 0.00797 (Fernandez et al., 2015a) calculated from the refractive index of $\eta =$

1.196 (Benson et al., 1983):



Atmospheric
Measurement
Techniques



Discussions

$$\gamma = \frac{(\eta - 1)^2}{(\eta + 1)^2} \tag{14}$$

As the liquid nitrogen in a container is covered with a styrofoam lid, the temperature of the cold target is given by the following equation with the transmittance factor of $t = 0.997$:

$$T_{cold} = t \cdot T_{LN2} + (1 - t)T_{amb} \tag{15}$$

### 3.2.2 Measurement

We measured the atmospheric spectrum at eight different zenith angles ranging from 44.46 degrees to 77.22 degrees with an integration time of 3.3 seconds for each measurement. Measurements at these angles are continuously repeated, including both hot and cold measurements. The spectra obtained from the same angle were averaged over 10-minute intervals, and these averaged spectra were treated as a single measurement set (top panel of Fig. 5). As shown in Fig. 5, the brightness temperature depends on the zenith angles because the air mass varies with the angle. The spectrometer provided spectral outputs from 16,384 channels with a resolution of 61 kHz. However, to reduce storage size, we performed binning of every five channels for the wing spectra during the data storage phase.



**Figure 5.** (Top) The ozone spectrum was measured at eight different zenith angles, ranging from 44.46 degrees to 77.22 degrees, over a 10-minute duration. The ten black asterisks (*) on both sides of the spectrum indicate the wing regions used to calculate tropospheric opacity, as described in Section 3.2.4. (Bottom) Opacity at ten frequencies (○) was calculated from the above spectrum, along with a linear approximation of opacity over the the entire frequency range (shown as a magenta line).



### 3.2.3 Pointing offset

The measurement of the atmospheric spectrum relies on the air mass as indicated in Eq. (5). Since the air mass is a function of the measurement zenith angle $\theta$, ensuring the accuracy of the zero-point of the zenith angle is important. Despite the presence of a level aligner on the axis of the mirror to set this zeropoint, an inherent limitation exists due to the step motor's minimum step angle of 0.18 degrees. However, we can use the Sun as a powerful point emitter for angle correction because the trajectory of the Sun is well-defined at any given geographic coordinate (Straub et al., 2011).

The SORAS mirror is oriented east-southeastward, so we have a chance to capture the solar emissions in the morning during certain periods. The available test period is constrained because the field of view is limited by both the physical window and the surrounding structures. Additionally, the test should be performed under a clear sky to minimize interference from water vapor and clouds. When the SORAS antenna is directed at the Sun through the opened window, there is a significant increase in measured brightness temperature. The brightness temperature, excluding solar influence, can be approximated using the radiative transfer equation with measured opacities $\tau_{tr}$ and the weighted mean tropospheric temperature $T_{trop}$ which will be discussed in the next section 3.2.4.

The sun scan measurements were conducted at 21 different angles within the zenith angle range of 65.52 to 78.48 degrees, with an observation time of approximately 2 seconds per angle. Including hot and cold measurements, each set comprised 23 angles in total. Considering the mirror rotation and delay times, the observation time for one complete set was 65 seconds. During the 45-minute observation period shown in Fig. 6, a total of 38 scans were performed. The opacity for $T_b$(RTE) was determined using the values observed immediately before the sunscan.

The difference between solar brightness temperature ($T_b$(Meas)) and estimated non-solar brightness temperature ($T_b$(RTE)) gives a solar radiation pattern shown in Fig. 6. This solar radiation pattern was interpolated across zenith offset angles at a specific azimuth direction, and then it was fitted to a Gaussian curve to determine the maximum presented in the left panel of Fig. 7. To determine the pointing offset, only the data of the $T_b$ difference larger than 25 K were considered. This analysis revealed a zenith pointing offset of 0.102 degrees (right panel of Fig. 7). Consequently, the zenith angle used in air mass calculations was corrected from the instrumental angle $\theta_{instr}$ to the real pointing angle $\theta_{SORAS}$ following Eq. (16).

$$\theta_{SORAS} = \theta_{instr} - 0.102 \tag{16}$$





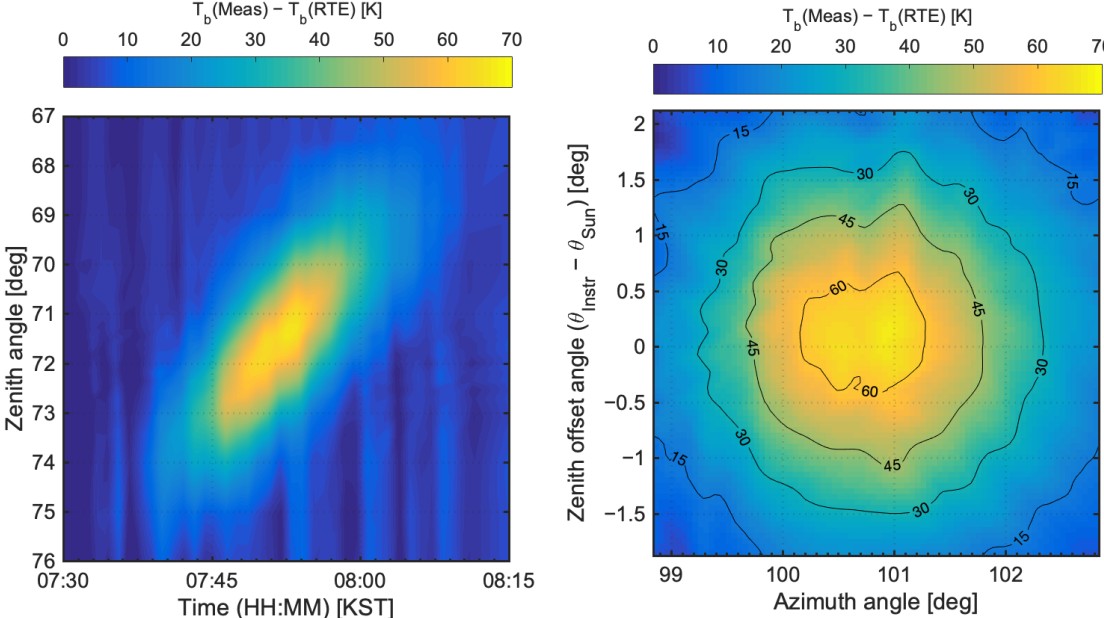

**Figure 6.** (Left) The difference in brightness temperatures observed during the Sun scan, illustrating the Sun's trajectory throughout the test. (Right) The results presented in the left panel have been transformed relative to the Sun's position.



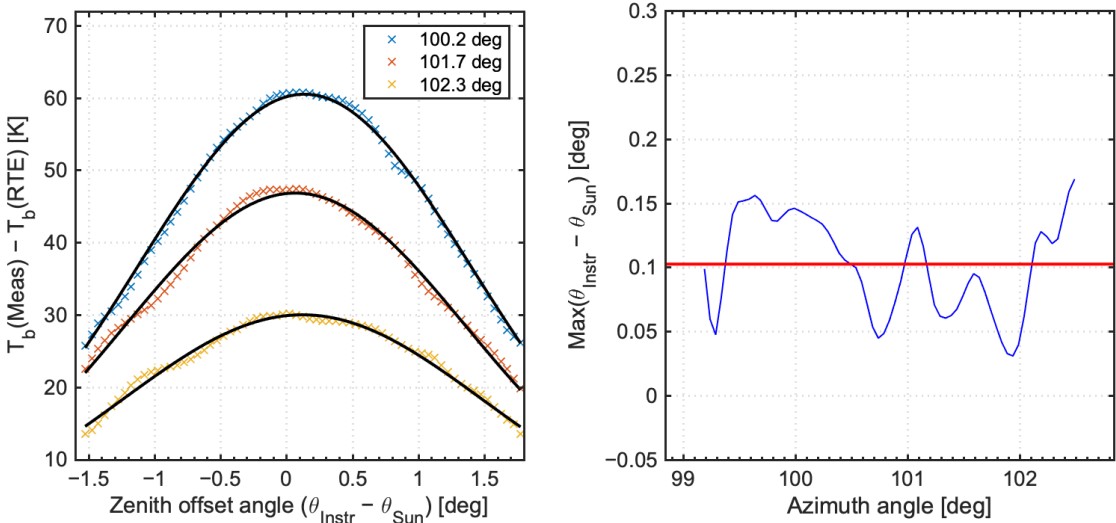

**Figure 7.** (Left) The difference in brightness temperatures at three different azimuth angles, with the Gaussian curve fit shown. (Right) The zenith offset angles from the azimuth view (in blue) alongside the averaged point offset of 0.102 degrees (in red).

### 3.2.4 Tropospheric opacity

The ozone spectrum $T_{b,O3}$ is reduced by the factor of $e^{-\tau_{tr}}$ as shown in Eq. (5). To restore $T_{b,O3}$ from the measured spectrum $T_b$, we need to estimate both tropospheric opacity $\tau_{tr}$ and the tropospheric temperature $T_{trop}$.

The calculation for $T_{trop}$ from Eq. (8) is possible when the vertical profile of the pressure, temperature, and humidity is given. However, at the time of observation, sufficient information was not accessible. In this study, $T_{trop}$ was determined by adjusting the ground temperature with a factor derived from climate data. Radiosonde data from the Osan station of the Korean Air Force (37.10 ºN, 127.03 ºE, about 50 km apart from the site) measured for 2 years between 2015 and 2016 was used to calculate $T_{trop}$ from Eq. (8). In Eq. (8), the tropopause height, $h$, was consistently set to 16 km throughout this period because the altitude of the observation platform in the retrieval process was fixed at 16 km. By assuming the linear relation between $T_{trop}$ and the ground temperature $T_{gr}$, we obtained the factor $\Delta T$ of –14.9 K at 110.4 GHz, which is applied for the overall frequency range of SORAS.

$$T_{trop} = T_{gr} + \Delta T \tag{17}$$

The estimated $T_{trop}$ from Eq. (17) is compared to $T_{trop}$ calculated with the vertical profiles of Korea Local Analysis and Prediction System (KLAPS) data provided by the KMA (top panel in Fig. 8). Both $T_{trop}$ variations are comparable with each





other; it turns out that the factor of –14.9 K is reasonable. It should be noted that the factor is dependent not only on the frequency but also on the local climate and the altitude of the site. The value of –14.9 K is specifically applicable to the 110 GHz at the SORAS location or a comparable site.

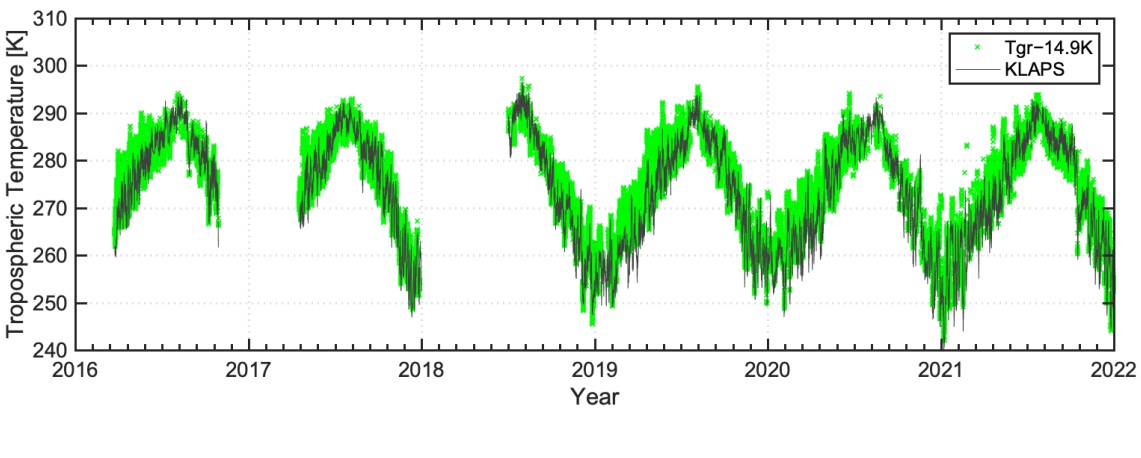

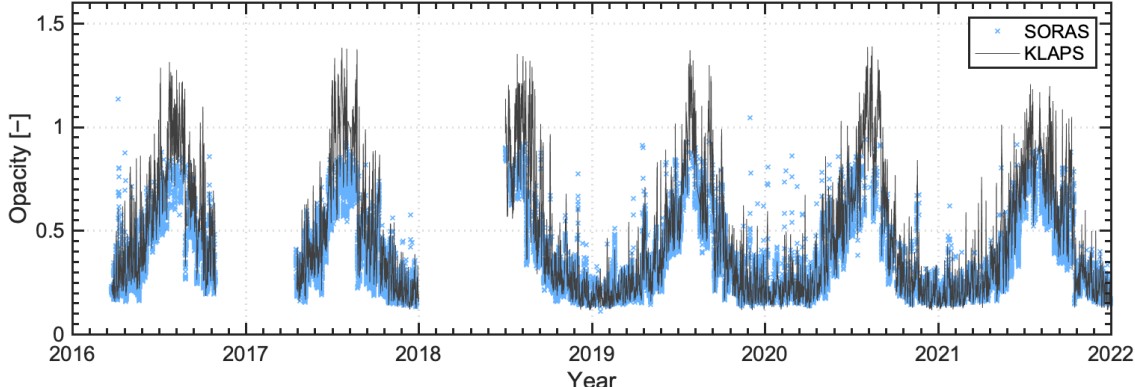

**Figure 8.** (Top) Tropospheric temperature at 110 GHz calculated from $T_{gr} - 14.9$ (green circle) and derived from Eq. (8) using KLAPS data at the SORAS location (black line). (Bottom) Tropospheric opacities at 110 GHz, measured by the tipping curve calibration (sky blue) and calculated using KLAPS data at the SORAS location (black line). Data gaps in 2016-2017 and 2018 are attributed to hardware maintenance and sight disturbances to the measurement view, respectively.

The tipping curve calibration (Han and Westwater, 2000), a general method to estimate the tropospheric opacity $\tau_{tr}$, is performed by observing the brightness temperature $T_b$ at various zenith angles $\theta$. When $T_b$ is measured away from the ozone transition frequency, it is reasonable to assume that $T_{b,O3}$ is approximately zero, allowing Eq. (5) to be rearranged as follows:

$$\tau_{tr} a_{tr}(\theta) = \ln\left(\frac{T_{trop} - T_{bg}}{T_{trop} - T_b(\theta)}\right) \tag{18}$$



Considering the non-ideal homogeneity of the atmosphere, it is essential to derive a converged opacity by discarding invalid data. Equation (18) is a linear function of $a_{tr}(\theta)$ and $\ln\left(\frac{T_{trop}-T_{bg}}{T_{trop}-T_b(\theta)}\right)$ with a slope $\tau_{tr}$, intersecting at the origin. The slope $\tau_{tr}$ is recalibrated after fitting the initial dataset. Subsequently, any data deviating from the established fit line are excluded based on the presumption that such deviations are induced by atmospheric inhomogeneity. This iterative procedure

ultimately yields a converged opacity value.

In this study, we measured atmospheric spectra at eight different zenith angles ranging from 44.46 degrees to 77.22 degrees. As shown in Fig. 5, the spectrum has a slope with respect to frequency, indicating that opacity varies with frequency within the SORAS observation range. To investigate this frequency dependence, five frequency regions on each side of the eight observed spectra were selected for opacity calculation. These frequencies were chosen from the wing regions, located

322 MHz to 382 MHz away from the ozone frequency of 110.836 GHz. The resulting opacities at 110.46 GHz are depicted in the bottom panel in Fig. 8, alongside calculated opacities from KLAPS data for comparison. The measured and the calculated opacities exhibit similar values, except during summer when spectra, even at different angles, are saturated due to high opacity, making it difficult to derive opacity from measurements. The obtained opacities on either side of the frequency were linearly fitted across the entire frequency range (bottom of Fig. 5).

Now, we can deduce the ozone spectrum $T_{b,O3}$ from the measured brightness spectrum $T_b$ using the tropospheric opacity $\tau$ and the tropospheric temperature $T_{trop}$. This results in an ozone spectrum (Eq. 19) that approximates measurements at the tropopause, even when excluding the cosmic background $T_{bg}$. This exclusion of $T_{bg}$ does not significantly impact the intensity or shape of $T_{b,O3}$, since $T_{bg}$ is assumed to be constant.

$$T_{b,O3} = \frac{1}{a_{mid}}\left\{\frac{(T_b-T_{trop})+(T_{trop}-T_{bg})e^{-\tau_{tr}a_{tr}}}{e^{-\tau_{tr}a_{tr}}}\right\} \tag{19}$$

## 295    4    Retrieval and validation

### 4.1    Spectrum setup

We acquired eight or fewer $T_{b,O3}$ measurements at different angles $\theta$ from a single set of observations, which are then averaged into a single spectrum for retrieval purposes. This approach aims to increase the number of measurements within a specific time frame and to minimize noise in the averaged spectrum. The retrievals were conducted using a 2-hour average of

$T_{b,O3}$ measured from April 2016 to December 2021. Subsequently, the spectrum was binned again into 5 channels for the center frequency area, akin to the wings, applying a frequency resolution of 305 kHz across the entire spectrum. The retrieval spectral bandwidth is set at 800 MHz, centered around 110.836 GHz with a ±400 MHz range. For the retrievals, only spectra with noise levels below 0.5 K are considered, which accounts for the reduced availability of profile data during summer.



As SORAS is oriented east-southeastward, a Doppler shift of the transition frequency 110.836 GHz may be induced by zonal winds in the upper stratosphere or mesosphere (Rüfenacht et al., 2012). Moreover, the exact frequency 110.83604 GHz as defined in the spectroscopic JPL catalog carries an uncertainty of 50 kHz. Errors in calculating the frequency corresponding to the spectrometer's channels can lead to the frequency offset. In this study, the frequency offset was determined by the curve fitting of $T_{b,O3}$. Figure 9 illustrates the procedure used to identify the frequency shift for $T_{b,O3}$.

Following the selection of the central area around 110.836 GHz $\pm$ 30 MHz (indicated in red), curve fitting of $T_{b,O3}$ versus the relative frequency in MHz is performed (indicated in black), as depicted in the middle and in the bottom panels of the figure. The offset is individually corrected prior to retrieval by adjusting the spectral frequency values. The average frequency offset observed from 2018 to 2021 was −55.1 kHz (Fig. 10), which is comparable to the 50 kHz uncertainty of the JPL catalog for 110.83604 GHz ozone transition frequency.

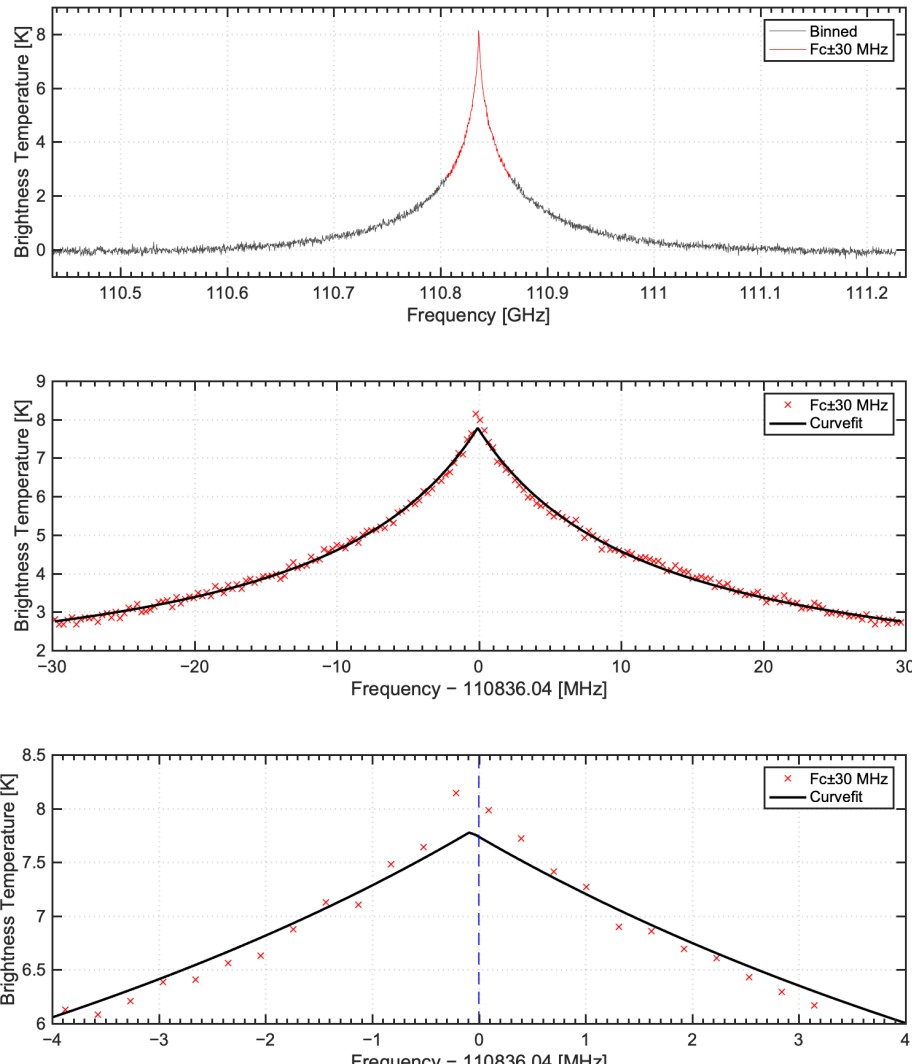

**Figure 9.** Procedure to determine the offset from the center frequency of the ozone spectrum. In this case, the frequency is shifted by −79.8 kHz from 110.83604 GHz.





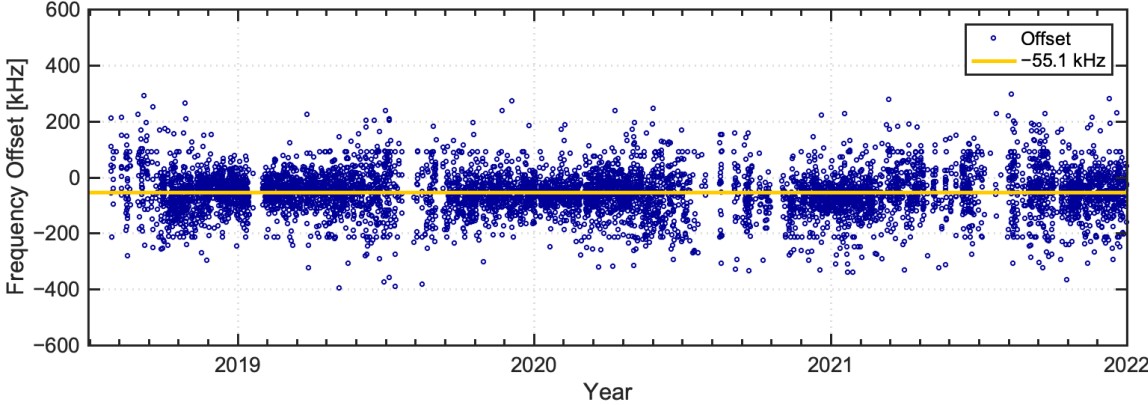

**Figure 10.** The individual frequency offset is determined by the curve fitting method. On average, the center frequency of the measurement is shifted by −55.1 kHz from 110.83604 GHz.

### 4.2 Retrieved ozone profile

The vertical profile of the stratospheric ozone is calculated through an optimal estimation method based on Bayes' theorem (Rodgers, 2000). The measured $T_{b,O3}$ is converted into the ozone profile for SORAS using the ARTS (v2.2)/Qpack2 software package (Eriksson et al., 2005, 2011). ARTS simulates the spectrum by combining the continuum and molecular transitions, utilizing atmospheric radiative transfer theory (the forward model) based on provided atmospheric profiles (also known as *a priori*). Qpack2 produces the optimal profile by comparing the simulated spectrum from ARTS with the measured spectrum

from SORAS, considering the uncertainties in both the measurements and a priori profiles.

The main parameters for the SORAS retrieval are listed in Table 1. Spectroscopic data are compiled by combining the ozone transition information from both the HITRAN and JPL catalogs. The a priori atmospheric profiles, including stratospheric ozone, are derived from climate monthly data near Seoul, measured by the AURA MLS satellite from 2006 to 2015. Spectral noise is assumed to be constant and is determined by the standard deviation within a 9 MHz wing portion of

335 the measured spectrum. The covariance matrix for the measurement is a diagonal matrix with elements equal to the square of the spectral noise. For the covariance matrix for the a priori profile, a Gaussian correlation function is employed between neighboring layers, setting the diagonal to a maximum value of 0.4 ppm from 10 hPa to 1 hPa. The baseline is approximated using a second-order polynomial function.

**Table 1.** Parameters utilized in the SORAS retrieval

| Center frequency | 110.83604 GHz |
|---|---|
| Spectroscopic catalog | JPL (for strength parameters) & HITRAN (for broadening parameters) |
| Atmospheric a priori | AURA MLS monthly averaged profile (2006–2015) |




| Frequency resolution | 305 kHz (5 channels binned) |
|---|---|
| Frequency bandwidth | 800 MHz (110.427 GHz ~ 111.227 GHz) |
| Spectrum noise level limit | ≤ 0.5 K |

Figure 11 shows an ozone spectrum and a simulated spectrum with its baseline. In the upper panel of Fig. 11, the simulated spectrum is about 1 K higher than the measured spectrum. This difference occurs bacause the measured spectrum was calculated using Eq. (19), where the opacity was determined based on the wings of the measured spectrum, causing

these wings to converge to zero. However, the actual spectrum observed at the tropopause retains a certain level due to cosmic background radiation and emissions from other atmospheric molecules. This effect is corrected by applying a baseline to the spectrum. The residuals, shown in the bottom panel, indicate no discernible pattern between the measurement and the simulation.

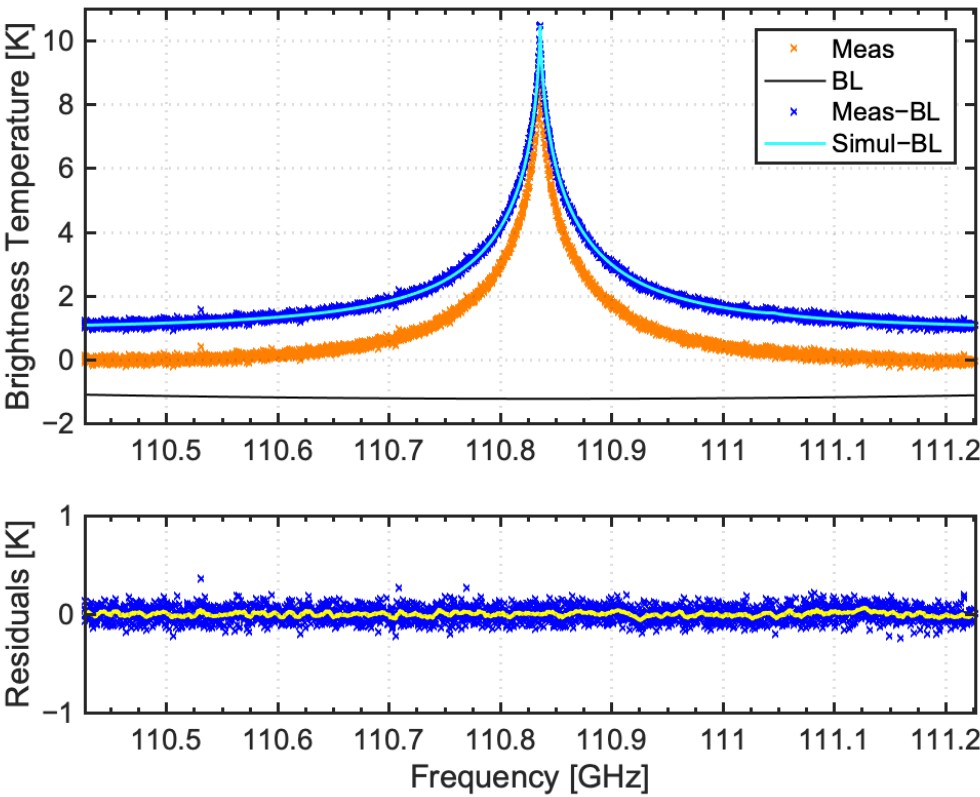

**Figure 11.** (Top) Ozone spectrum measured over 2 hours (orange and blue) alongside the simulated spectrum (cyan), with a second-order polynomial baseline (black). The measured spectrum after baseline correction is displayed in blue. (Bottom) Residuals between measured and simulated spectra, with a smoothed line for clarity.





The retrieved ozone profile, along with the a priori profile and MLS profiles, is shown in the two left panels of Fig. 12. The total retrieved error is indicated by the yellow-shaded area. To account for the differing resolutions between the measurements and the MLS data, the MLS data were convolved with the averaging kernel matrix and the a priori profile of SORAS (Tsou et al., 1995).

$$x_{MLSConv} = x_a + A(x_{MLS} - x_a) \tag{20}$$

where $x_{MLSConv}$ is the convolved profile, $x_a$ is a priori profile, $A$ is averaging kernels, and $x_{MLS}$ is the higher-resolution MLS profile. The third panel of Fig. 12 illustrates the smoothing and measurement errors, along with the total error, calculated as the square root of the sum of the squares of each individual error. In this case, smoothing error dominates the total uncertainty across the entire altitude range.

The averaging kernels and the measurement response are displayed in the right panel of Fig. 12. The altitude where the measurement response exceeds 0.8 is indicated in gray-shaded color. The measurement response quantifies the contribution of the actual measurements to the retrieved profile, as compared to the a priori profile. It is inversely proportional to the measurement noise, which is limited to a maximum of 0.5 K in our case. During the winter, when measurement noise falls below 0.1 K, the altitude range with a measurement contribution greater than 80% extends from 60.8 hPa to 0.08 hPa (approximately 19 km to 65 km). However, in the summer season, the altitude range narrows to between 43.7 hPa to 0.7 hPa (approximately 21 km to 50 km), coinciding with noise levels exceeding 0.3 K. Figure 13 presents the ozone profiles measured by SORAS in Seoul since 2016.



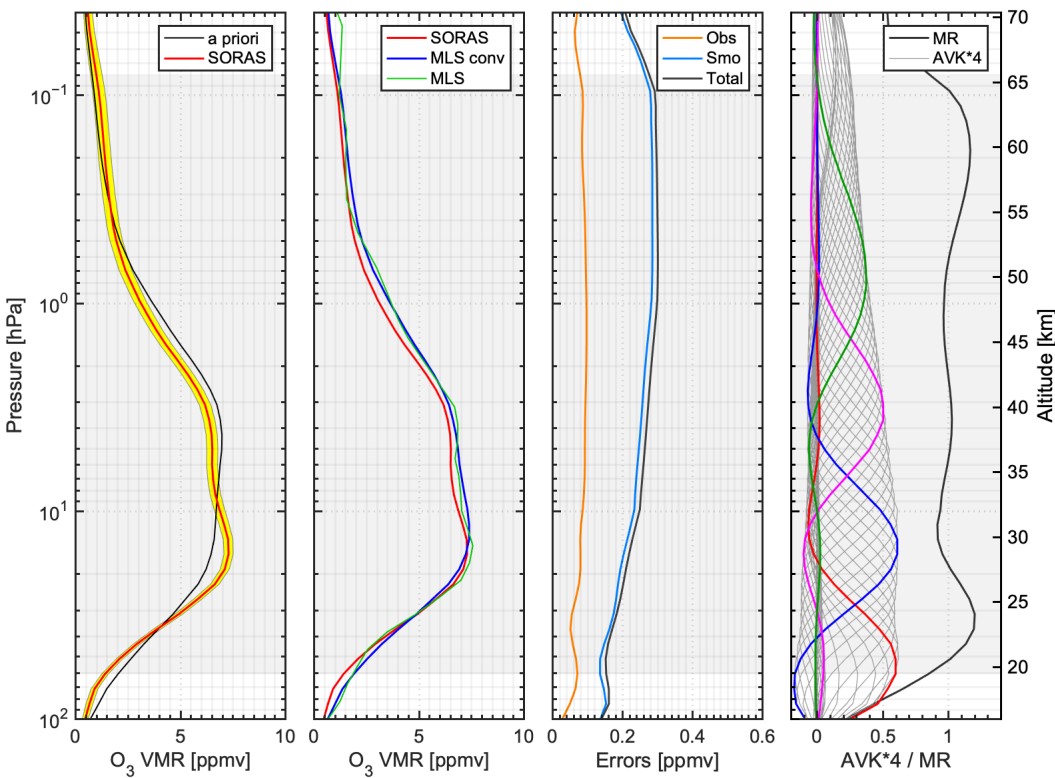

**Figure 12.** Ozone retrieved profile at 18:00 (UTC), 14. Dec. 2019 is presented alongside an a priori profile and total error limits. The second panel compares the MLS convolved and the original MLS profile. The third panel describes the total error, including both smoothing and observation errors. In the right panel, the averaging kernels and the measurement responses are displayed; altitudes where the measurement response exceeds 0.8 are shaded in gray. The averaging kernels at 20, 30, 40, and 50 km are presented with thick colored lines.





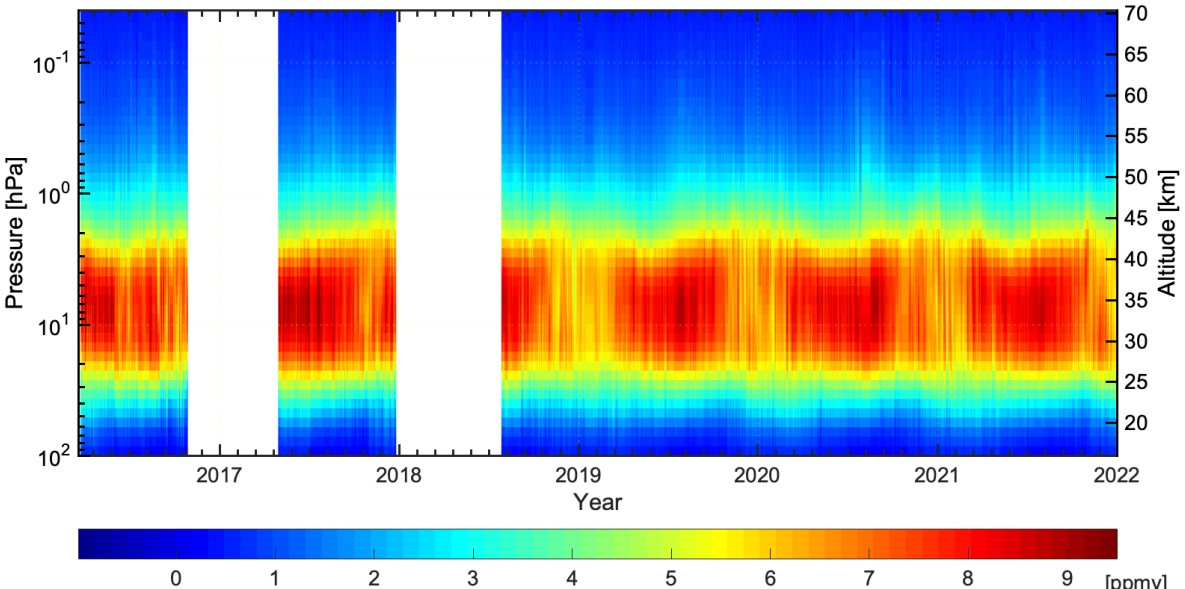

**Figure 13.** The SORAS microwave radiometer measured the ozone profile above Seoul.

Figure 14 shows the ozone volume mixing ratio measured by SORAS alongside the MLS convolved profile across four pressure intervals. The SORAS data times are confined to within ±1 hour of the MLS measurement. The difference between the two datasets is shown in Fig. 15. For most of the period, excluding summer (June to September), there is a negative bias. When comparing the opacity between SORAS and KLAPS in Fig. 8, it is clear that the opacity of SORAS during summer is lower than that of KLAPS. This difference affects the ozone spectrum $T_{b,O3}$ calculation in Eq. (19), resulting in a different bias compared to other periods. Outside of summer, the mean biases are −0.21 ppmv (−8.8 %) at altitudes of 16~24 km, −0.27 ppmv (−4.0 %) at 24~34 km, −0.64 ppmv (−9.8 %) at 34~46 km, and −0.41 ppmv (−15.1 %) at 46~55 km. According to Sauvageat et al. (2021), these systematic biases may be attributed to unexplained spectral leakage in the AC240 spectrometer. The paper also presents ozone concentration biases at altitudes of 20~70 km, measured using the AC240 and the recent U5303 spectrometer, which range from −6% to −11%. These values are comparable to the biases observed in the SORAS data.



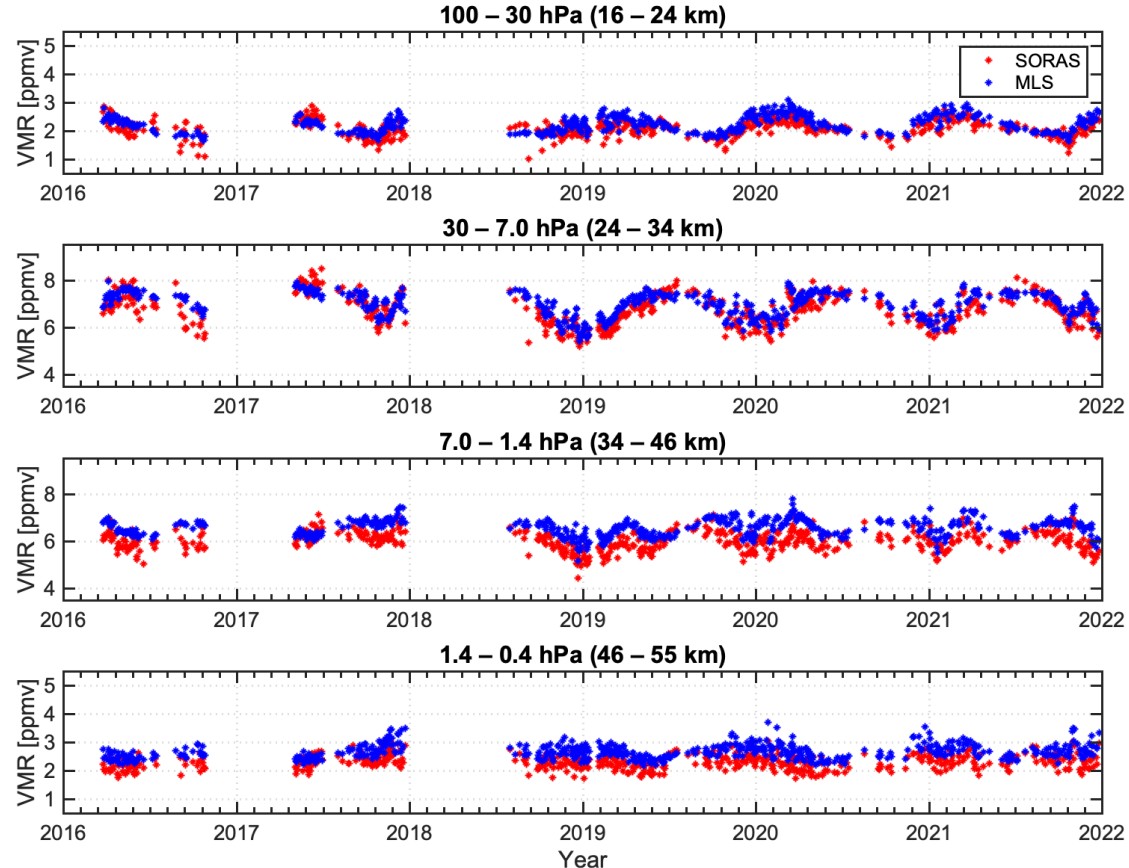

**Figure 14.** The ozone volume mixing ratio above Seoul by SORAS (red) and MLS (blue) at four pressure levels.

395



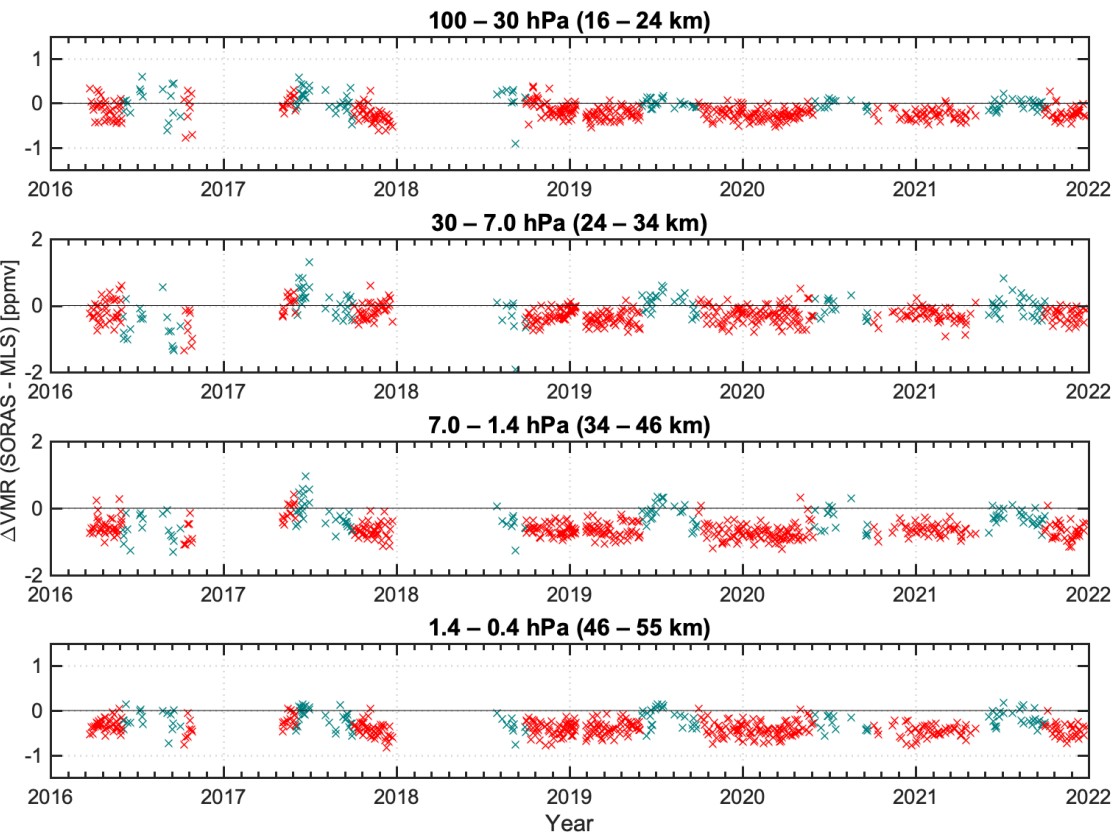

**Figure 15.** Difference in ozone volume mixing ratio between SORAS ans MLS convolved profiles. Red marks (×) represent data observed from January to May and October to December.

## 5   Conclusions

SORAS is a 110.836 GHz ground-based radiometer for monitoring the vertical profiles of stratospheric ozone in Seoul. It is the first microwave radiometer developed in Korea to measure stratospheric ozone's vertical structure. The radiometer is a heterodyne receiver, featuring a corrugated horn antenna with a full width at half maximum of 8.3 degrees. The microwave components allow the incident radiation to be analyzed by a digital FFT spectrometer. The ozone spectrum is measured using hot-cold calibration and continuous tipping curve calibration at eight different zenith angles to estimate the tropospheric effect. The pointing offset, determined by a Sun-scanning method, is 0.102 degrees. The weighted mean tropospheric temperature at 110 GHz is obtained by subtracting 14.9 K from the measured ground temperature.



The tropospheric contribution to the measured spectrum is eliminated prior to the retrieval. The frequency offset from the ozone transition frequency of 110.83604 GHz may be induced by a Doppler shift, spectroscopical, or instrumental error. This is corrected by fitting the curve of the spectrum and aligning the center frequency with 110.83604 GHz. The average offset is determined to be –55.1 kHz. The retrieval was performed through an optimal estimation method from SORAS measurements between 2016 and 2021. The retrieved ozone profiles above Seoul are described in this paper and compared with AURA MLS convolved profiles. Excluding the summer, SORAS ozone profiles show lower values compared to MLS profiles, with differences ranging from –4% to –15% depending on the altitude. This negative bias is considered to originate from the AC240 spectrometer.

***Data availability.*** Data can be made available by contacting the corresponding author.

***Author contributions.*** SK conducted the SORAS operation and data analysis and wrote the manuscript. JO was responsible for the development of the SORAS and provided overall guidance and supervision for the study.

***Competing interests.*** The contact author has declared that none of the authors has any competing interests.

***Acknowledgments.*** This study was supported by the Korea Meteorological Administration Research and Development Program under Grant KMI2021-02510. The authors would like to acknowledge to Dr. Se-Hyung Cho from the Korea Astronomy and Space Science Institute for providing the motivation for the development of SORAS. Additionally, special thanks go to Dr. Prof. Niklaus Kämpfer and Dr. Axel Murk from the University of Bern, Switzerland, for their assistance in optimizing SORAS.

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
