# Peer review of "SORAS, A ground-based 110 GHz microwave radiometer for measuring the stratospheric ozone vertical profile in Seoul"

_Atmospheric Measurement Techniques, 2024_

## Referee Comment (RC2)

BLOCK DIAGRAM SYMBOLS FOR FILTERS

---

## Author Comment (AC1)

**Response to Reviewers' Comments on the preprint amt-2024-108**

Dear Reviewers,

We greatly appreciate your valuable and insightful comments. We have carefully considered your suggestions and revised the manuscript to address the points raised. Your feedback has significantly improved the clarity and rigor of our work.

We have provided our responses below. Your comments are enclosed in boxes, and our corresponding responses are written below each box. The line numbers of the revisions have been indicated for both the revised manuscript (RM) and the tracked changes file (TCF).
* * *
**RC1: 'Comment on amt-2024-108', Anonymous Referee #2, 24 Sep 2024**

> This manuscript describes a microwave radiometer that has been measuring ozone profiles from Seoul since 2016. The authors do a nice job addressing details of the measurements that are extremely important in obtaining accurate retrievals (e.g. tropospheric opacity and pointing). Understanding such details is important for any group attempting to make such measurements. I thank the authors for addressing my concerns in the initial review.
>
> 1) Figure 8, line 287, and conclusion – Are the tropospheric opacities shown in Figure 8 for KLAPS and SORAS calculated at precisely the same time? It is not clear whether the higher summer opacity of the KLAPS results come from periods when SORAS opacity measurements are not possible because of the high opacity. In the conclusion the authors say that the O3 bias relative to MLS is different during the summer than during other seasons, but in order to establish this a clear plot showing the difference in tropospheric opacity biases is needed.

[in RM: Line 403 and Fig. 16  //  in TCF: Line 431 and Fig. 16]

The opacity shown in Figure 8 for SORAS and KLAPS represents the data available for the same period. SORAS spectra were averaged at 10-minute intervals plotted accordingly, while KLAPS data were provided at a 1-hour interval. To compare the two datasets, SORAS data corresponding to the KLAPS time were extracted, and a plot showing the differences between the two datasets is provided below. This plot is intended to be added as Figure 16 in the manuscript. The opacity of SORAS during summer is lower than that of KLAPS, as shown in Fig. 16. The low calculated opacity affected the intensity by overestimating the contribution of $T_{b,O3}$ in the observed spectrum $T_b$. As a result, the comparison between SORAS and MLS led to a different bias during the summer compared to other seasons.

[Figure]

Figure 1. Difference in opacity between SORAS and KLAPS (see Fig. 8). Only SORAS data matching the time of the KLAPS data were used. The red marks are indicated as described in Fig. 15.

2) Paragraph starting on line 382 and conclusion - While the authors are correct in attributing a portion of the observed negative bias to the use of an AC240 spectrometer, this is not thought to cause an altitude dependence in the bias. Just as for the instrument presented here, the Sauvegeat study does show an altitude dependence in the bias between that ground-based instrument and MLS. But, as for the instrument discussed in the Sauvegeat study, the altitude dependence of that bias is only weakly, if at all, caused by the AC240 (their Table II, last column). This should be noted in the text.

[in RM: Line 410 and 436 // in TCF: Line 439 and 468]

We addressed it to the manuscript.

3) Lines 305 and 409 – What velocity does a 55 kHz doppler shift imply? Is this physically realistic? Is the 55 kHz smaller than the expected error of the local oscillators used in the instrument?

[in RM: Section 4.1 Spectrum setup and 5. Conclusions]

The 55 kHz Doppler shift corresponds to a velocity of 149 m/s, which is significantly higher than the known wind speeds. Another reviewer raised a similar concern regarding this issue, prompting us to re-examine the matter carefully. Upon recalculating, we discovered that the frequency error increased to –86 kHz due to incorrect mapping between the spectrometer channels and frequencies. Additionally, we identified that the local oscillator within the baseband converter has a –6 kHz offset.

Nevertheless, we observed notable frequency offsets of –80 kHz, as well as high occurrences at –245 kHz and –60 kHz. These offsets appear to result from instabilities in the local oscillator, although the exact cause remains unclear. We have updated the manuscript to include these findings, and texts to the Doppler shift and JPL frequency error has been revised.

Our response about this issue overlaps with the points addressed in item 12 of Referee #3's comments. The content has been provided below.

We identified that the offset is not caused by the Doppler effect but instead originates from an incorrect channel definition in the AC240 spectrometer and a frequency offset in the local oscillator.

According to Benz et al. (2005), the AC240 spectrometer analyzes a 1 GHz bandwidth using 16384 channels. As stated in Eq. (2) of the reference, the center frequency ($\nu$) corresponding to a channel is calculated as:

$$\nu_{AC240} = \frac{m}{16384} + \frac{0.5}{16384} \quad (m = 0, 1, 2, \ldots, 16383)$$

where $m$ represents the channels for positive frequencies and ranges from 0 to 16383. As the frequency in Eq. (2) was defined as the lower edge of channel $m$, 0.5/16384 was added to represent the center frequency of the channel. However, in this study, the first channel of the spectrometer was incorrectly defined as 1 instead of 0, and the frequency was calculated by dividing the channel number, $k$, by 16384. Therefore, the corrected frequency at the spectrometer of SORAS should be calculated as follows:

$$\nu_{soras} = \frac{k-1}{16384} + \frac{0.5}{16384} = \frac{k-0.5}{16384} \quad (k = 1, 2, 3, \ldots, 16384)$$

The corrected frequency is 30.5 kHz, smaller than the incorrectly defined frequency, and Fig. 9 and Fig. 10 have been revised to reflect the new frequency.

This adjustment further increased the offset from $-55.1$ kHz to $-85.6$ kHz.

Following your suggestion, we also plotted a histogram of the offsets. The histogram shows symmetry around the mean offset of −86 kHz. However, there is a significant frequency of offsets at −245 kHz and 60 kHz. As you mentioned, this appears to result from frequency shifts to specific levels caused by instability in the local oscillator, although it is unclear at which stage of the frequency conversion this occurs. Additionally, as you pointed out, such LO frequency shifts are likely to have a significant impact when averaging long-term observation data.

[Figure]

Figure 2. (Left) The individual frequency offset is determined by the curve fitting method. On average, the center frequency of the measurement is shifted by −85.6 kHz from 110.836 GHz. (Right) The frequency offset distribution is presented as a histogram, showing high occurrences of frequency shifts at −245 kHz and 60 kHz.

We examined the local oscillator of the baseband converter. The LO frequency has a frequency offset of −6 kHz from 2 GHz. The −6 kHz LO offset contributes to a −85.6 kHz shift.

We have incorporated it into the manuscript.

The content in the manuscript referring to the Doppler effect has been removed, and the text was revised to prevent further misunderstanding. Unlike previous studies investigating the Doppler effect with radiometers, such as Rufenacht et al. (2012) (6.1 kHz) and Hagen et al. (2018) (12.2 kHz), the frequency resolution in this study is 5 to 10 times larger. Additionally, due to the lack of opposite-direction observation corrections, the observation conditions in this study are insufficient to evaluate the Doppler effect.
* * *
4) Line 343 – 'bacause' should be 'because'

[in RM: Line 360  //  in TCF: Line 388]

We corrected it.

---

## Author Comment (AC2)

**Response to Reviewers' Comments on the preprint amt-2024-108**

Dear Reviewers,

We greatly appreciate your valuable and insightful comments. We have carefully considered your suggestions and revised the manuscript to address the points raised. Your feedback has significantly improved the clarity and rigor of our work.

We have provided our responses below. Your comments are enclosed in boxes, and our corresponding responses are written below each box. The line numbers of the revisions have been indicated for both the revised manuscript (RM) and the tracked changes file (TCF).
* * *
**RC2: 'Comment on amt-2024-108', Anonymous Referee #3, 06 Dec 2024**

> The authors introduce the South Korean millimeter wave radiometer SORAS (Stratospheric Ozone RAdiometer in Seoul), designed to monitor the ozone emission line at 110.8 GHz. In its original version from 2008 SORAS has been a double sideband radiometer. In 2016 it has undergone a transition into a single sideband radiometer. The changes made to the original SORAS are unclear as the three references to earlier studies with SORAS are published in Korean language, indecipherable to me (Line 59). However, the manuscript presents measurements and results of the post-2016 period, which makes references to the older studies between 2009 and 2014 not necessarily needed.

[in RM and TCF: Line 59]

We have removed references to older studies.

> Today SORAS is one of the two Asian radiometers currently in operation monitoring stratospheric ozone, which makes it a relevant instrument, even though neither technique, idea nor measurement processes are new. However, in SORAS there has been implemented the concept of direct amplification together with a high pass filter blocking the contribution from the lower (image) sideband, which avoids complications due to a quasi-optical single-sideband filter.
>
> The description of the instrument, the calibration method and the data acquisition are described well, probably even too much in detail given the large number of references where these equations are presented. However, the methodology is sound and has proven successful with other radiometers as well. The authors address certain instrumental imperfections and their corrections.
>
> The language is fluent and the authors give proper credit to related work.
> The abstract provides a quite concise and complete summary and the overall presentation is well structured and clear.
>
> In the end the authors present results (Fig. 13) which show a stable and reliable observation period over roughly six years which makes SORAS a valuable source of ozone data from a region that has rather small contribution to the overall knowledge of the state of the ozone layer.
>
> In order to support and strengthen their results the authors validated their data with the 'gold standard' of atmospheric remote sensing, MLS. This comparison shows a nice agreement between data from SORAS and MLS, after MLS data altitude resolution has been adopted to SORAS with the help of SORAS' averaging kernels. A slight bias is attributed to the AC240 spectrometer.

The number and quality of references are quite appropriate. I would suggest adding the Parrish et al. (1988) as a very basic reference to ground-based microwave radiometry.

I have not seen any supplementary material.

[in RM and TCF: Line 34]

We have added the reference, Parrish et al. (1988).

Specific questions and comments:

1) Line 138: The authors write 'As the tropospheric ozone is present in very small amounts, its contribution to $T$b can be negligible.' Can the ozon below 20 km really be neglected when the AKs for the 20 km altitude in Fig. 12 shows substantial sensitivity even below 20 km and the SORAS altitude range is estimated to 16 to 70 km. Elaborate!

[in RM: Line 140  //  in TCF: Line 146]

Figure 12 in the manuscript provides an example of retrieval, with the altitude corresponding to an 80% measurement response (MR) shaded in gray. This highlights that below 20 km, the measurement response decreases, indicating that the influence of the a priori becomes more significant than that of the actual observed spectrum.

The mention of ignoring tropospheric ozone in the manuscript indicates that the ozone concentration distributed in the troposphere is less than 100 ppbv, which is much smaller than the stratospheric ozone concentration measured in ppmv. Separating the ozone signal from the continuum becomes challenging due to the low concentration and pressure broadening. The figure below simulates the ozone spectrum observed from the ground, assuming an ozone concentration of 70 ppbv below 16 km in a typical ozone profile without considering the influence of the continuum. Each spectrum corresponds to ozone concentrations up to 70 km, 50 km, 30 km, 20 km, and 16 km, respectively. The figure shows that the ozone signal diminishes and broadens as the maximum altitude decreases. The signal is extremely weak for ozone limited to 16 km, below 0.07 K, which is negligible. We replaced $T_b$ with $T_{b,O3}$ in the line 139 sentence to make it clear.

[Figure]

Figure 1. (Left) Ozone profile used in the spectrum simulation. (Middle) Simulated ozone spectrum considering ozone profile from the surface to the upper altitude limit. (Right) Magnified view of the spectrum."

2) From Fig. 2 and paragraph 3.2.1: I would suggest the authors use more common symbols for high pass and band pass filter in the block diagram and explain the symbol with the two arrows of which the lower one is crossed. I apologize if this is just my limited knowledge that lead to this comment but at least for the filters I am used to present the symbols as shown in the attached pdf document . (https://www.hobbyprojects.com/general_theory/filters.html)

> I would also avoid the mentioning of upper and lower sideband, USB and LSB, in the sketch as output from the mixer, since the mixer output inherently usually contains both sidebands, even if the RF signal INTO the mixer might have a suppressed unwanted sideband. I would suggest writing 'IF' for the intermediate frequency as the output of the mixers instead.
>
> Has the effect of Styrofoam on the measurements been investigated under different weather conditions and over a longer period? Does it take into consideration that water vapor could penetrate into the material? If so, looking through the Styrofoam at different angles might lead to a varying contribution to the measured brightness temperature.

[in RM and TCF: Fig. 2]

As you suggested, the filter symbol in Figure 2 has been revised, and the USB and LSB labels have been updated to IF.

We appreciate your comment regarding the potential impact of Styrofoam on transmittance during humid weather. Considering the high humidity in Korean summers and the discrepancies observed in this study between the summer opacity and retrieval results compared to reference values, it seems worthwhile to investigate the effects of moisture absorption on Styrofoam's transmittance. Unfortunately, we do not have data from past observation periods to evaluate such changes. We will refine your suggestion further and explore ways to incorporate it into future observations. Thank you again for your feedback.

> 3) Line 94: Frequencies lower than 100 GHz are cut off …

[ in RM: Line 93 // in TCF: Line 98]

Based on your suggestion, the mixer's output is influenced by both sidebands, so we have deleted the relevant sentence.

> 4) Line 98: The baseband converter with a LO frequency of 2.0 GHz produces an output of 0.609 GHz +/- 415 MHz I assume. What are the numbers in the sketch (1070~1900 MHz) telling the reader?

[in RM: Line 97 // in TCF: Line 103]

The baseband converter was designed with a 1500 MHz input frequency, a 2 GHz LO frequency, and an asymmetric bandwidth of 830 MHz. Based on this bandwidth, the valid input frequency range for the baseband converter is 1070 MHz to 1900 MHz. However, the frequency range for the BBC, as you pointed out, has been removed since the usable band of 100–930 MHz is indicated at the final stage of the diagram. To ensure clarity, the frequency range for the 830 MHz bandwidth has been added to the manuscript.

> 5) Line 101 What does the sentence mean that starts with 'As a result …'? Does it state the NOMINAL frequency range of the baseband converter should be 110.327 to 111.157 GHz, BUT it turned out that the ACTUAL frequency range of the baseband converter is somewhat shifted? And thus, the spectrum started at higher channels within the FFTS? Even a shift of 100 MHz should keep the entire spectrum well inside the FFTS bandwidth of 1 GHz. For reasons of clarity I would suggest presenting one example spectrum as measured by the 1 GHz bandwidth FFTS.

[in RM: Line 102-105 // in TCF: Line 108-111]

This content is linked to the aforementioned BBC bandwidth of 1070 – 1900 MHz. While the FFT of SORAS has a 1 GHz bandwidth, the effective frequency range is determined by the 830 MHz bandwidth of the BBC. Therefore, the RF frequency range usable for this study is limited to 110.327 GHz – 111.157 GHz of the 830 MHz bandwidth. However, upon examining the observed spectrum within this range, unidentified signals were detected below 110.4 GHz, and this region was excluded from the analysis. Additionally, frequencies above

111.157 GHz appeared usable despite the constraints of the BBC bandwidth. To ensure the spectrum is symmetrically centered around 110.836 GHz, the spectral frequency range used in this study was defined as 110.427 GHz – 111.227 GHz.

The full spectrum used for observations is shown below. Considering the 1 GHz bandwidth of the FFT spectrometer, the full spectrum should be plotted for the frequency range of 110.227 GHz to 111.227 GHz. For the data storage, frequencies below 110.319 GHz were not retained in order to save the storage, meaning the full 1 GHz bandwidth is not represented.

[Figure]

Figure 1. The SORAS full spectrum ranging from 110.319 GHz to 111.227 GHz.

The comment made us realize that an explanation for the shift in the starting frequency from 110.327 GHz to 110.427 GHz was missing. We have now added this explanation to the manuscript.

6) Fig 6. (Right) I would suggest 'Sun azimuth angle' for clarity reasons. How was the zenith of the sun calculated? As the sun was rather low above the horizon, was the light refracting property of the atmosphere included into the calculations? The sun might have appeared higher than it actually was and the angle correction needed would be even larger.

[in RM: Fig. 6, 7 and Line 221  //  in TCF: Fig. 6, 7 and Line 227]

We updated the 'azimuth angle' to 'Sun azimuth angle' for Fig. 6 and Fig. 7 and add Duffett-Smith et al. (2011) reference.

The method for calculating the Sun's trajectory is well described in Duffett-Smith et al. (2011), while the MATLAB code for trajectory calculation and the Sun scan application method are detailed in Straub et al. (2011). The specific equations are also provided in Duffett-Smith et al. (2011), which has been added as a reference in the manuscript.

Below, we have briefly summarized the methods for calculating the Sun's altitude and azimuth, along with the relevant sections from the references. Most of the symbols used in the equations are consistent with those in the references.

| Method | Section of the Duffett-Smith et al. (2011) |
|---|---|
| 1. Calculate the Julian date ($\boldsymbol{JD}$) | Section 4 |
| 2. Calcualate the ecliptic longitude of the Sun ($\boldsymbol{\lambda}$) | Section 46 |

| | | |
|---|---|---|
| $$\lambda = \frac{360}{365.242191}D + \frac{360}{\pi}e\sin\left(\frac{360}{365.242191}D + \epsilon_g + \omega_g\right) + \epsilon_g$$
 $e$: the eccentricity of the Sun-Earth orbit
 $\epsilon_g$: the Sun's mean ecliptic longitude at the epoch
 $\omega_g$: the longitude of the Sun at perigee
 $D$: the number of days since the epoch | | |
| 3. Calculate the obliquity of the ecliptic ($\epsilon$)
 $$\epsilon = 23.439292 - (46.815x + 0.0006x^2 + 0.00181x^3)/3600$$
 $$x = \frac{JD - 2451546}{36525}$$ | Section 27 | |
| 4. Calculate the declination ($\delta$)
 $$\delta = \sin^{-1}(\sin\beta\cos\epsilon + \cos\beta\sin\epsilon\sin\lambda)$$
 $$\approx \sin^{-1}(\cos\beta\sin\epsilon\sin\lambda)$$
 $\beta \approx 0$ for the Sun | Section 27 | |
| 5. Calculate the right ascension ($r$)
 $$r = \tan^{-1}\left(\frac{\sin\lambda\cos\epsilon - \tan\beta\sin\epsilon}{\cos\lambda}\right) \approx \tan^{-1}\left(\frac{\sin\lambda\cos\epsilon}{\cos\lambda}\right)$$ | Section 27 | |
| 6. Calculate the hour angle ($H$)
 $$H = LST - r$$
 $LST$: Local sidereal time
 $LST$ calculation can be performed by converting Universal Time ($UT$) to Greenwich Sidereal Time ($GST$). Detailed information can be found in sections 12 and 14 of the reference book. | Section 24
 Section 12
 Section 14 | |
| 7. Calculate the true altitude ($a$)
 $$a = \sin^{-1}(\sin\delta\sin\phi + \cos\delta\cos\phi\cos H)$$
 $\phi$: the observer's geographical latitude | Section 25 | |
| 8. Calculate the azimuth ($A'$)
 $$A' = \tan^{-1}\left(\frac{-\cos\delta\cos\phi\sin H}{\sin\delta - \sin\phi\sin a}\right)$$ | Section 25 | |
| 9. Calculate the refraction ($R$)
 $$R = \frac{P(0.1594 + 0.0196a + 0.00002a^2)}{T(1 + 0.505a + 0.0845a^2)}$$
 $P$: barometric pressure in hPa
 $T$: temperature in K | Section 37 | |
| 10. Calculate the apparent altitude considering refraction ($a'$)
 $$a' = a + R$$
 (for the zenith angle $\theta = 90 - a'$) | Section 37 | |

[Reference]

Duffett-Smith, P., Zwart, J., and Duffett-Smith, P.: Practical astronomy with your calculator or spreadsheet, 4th ed., Cambridge University Press, Cambridge; New York, 2011.

Straub, C., Tschanz, B., Murk, A., and Kämpfer, N.: Scanning the Sun to determine the pointing of a 22 GHz water vapor radiometer, 2011-04-MW, Institute of Applied Physics, University of Bern, 2011.

| |
|---|
| 7) Line 130 I suggest to mention the condition for which the Rayleigh-Jeans approximation can be used:
 $h\nu \ll kBT$ |

[in RM: Line 131 // in TCF: Line 137]

We added the relevant condition for the Rayleigh-Jeans approximation.

8) Line 140, Eqs. (5-7): It seems unfortunate to me that the authors use the variables '$a$' (italic a) and '$a$'(italic alpha) for air mass and absorption coefficient respectively in the same paragraph 3.1. The italic '$a$' is hardly distinguishable from the Greek 'alpha'. I suggest using '$A$' as variable for the air mass, as for instance Parrish et al. (1988).

: The symbol for air mass has been changed from '$a$' to '$A$' throughout the manuscript.

9) Line 143 As a matter of taste, I leave it to the authors to decide whether the issue of the air mass factor has to be explained in such detail or rather referred to any of the references.

: The definition of air mass is widely known, and simplified methods such as $1/\cos\theta$ are often used to calculate it. In this study, however, we wanted to emphasize that a method accounting for the Earth's curvature was applied, and the tropopause height was set to 16 km. Therefore, we prefer to keep this content as it is.

10) Line 184 and Eq. (12): How confident are the air temperature measurements on the roof of the building? Solar radiation at low latitudes can lead to increased air temperature due to convection on the roof, with rather strong diurnal variation. Other sources, such as Parrish et al., (1988) assume 7K as a reasonable temperature correction, so 14.9 K seems a rather large number, even if Ingold et al. suggest 10 - 20 K in Ingold.

: Thank you for your comment regarding the potential influence of convection as an explanation for the 14.9 K difference, which deviates significantly from other data. In this study, $T_{trop}$ was calculated using Eq. (8), where the absorption coefficient varies depending on frequency, temperature, and pressure. In the case of Parrish et al. (1988), the measurement frequency range is between 260 GHz and 280 GHz, which is different from the 110 GHz range used in this study, and the altitude of the site is 4300 m, much higher than the 52 m used in our case. Different climatic conditions could also be a contributing factor. Additionally, Parrish's paper employs a different approach to calculating $T_{trop}$ compared to our study. For these reasons, the factor used in the $T_{trop}$ calculation is expected to differ. The weather sensor was installed at a high location on the roof to minimize the heat effect from the ground. However, no comparable data are available to evaluate convection.

[Figure]

Figure 2.  Installation of the weather sensor (WXT510) on the roof of the building

11)  Line 256: which observation platform?

[in RM: Line 259  //  in TCF: Line 271]

The influence from the atmosphere below 16 km was treated as a continuum and excluded from the ozone signal in Eq. (19), suggesting that the radiometer was configured to measure observations as if it were at 16 km altitude. However, since the tropopause height was already defined as 16 km and the relevant details were explained alongside Eq. (19), this phrase has been deleted.

12)  Line 295 and paragraph 4.1 and Fig. 10 : This paragraph is of larger concern and the problem of frequency fluctuations needs more attention. The average 55 kHz frequency difference between the measured center frequency and the one in the JPL catalogue is a large deviation from the catalogue value. Moreover the spread in frequency offset (± 200 kHz) as shown in Fig. 10 is surprisingly large and cannot by any means be explained by Doppler effect. Already a 50 kHz offset would require wind speed of roughly 500 km/h in the stratosphere. In Fig 10 there are deviations of hundreds of kHz from the catalogue value, leading to 3000 km/h for an offset of 300 kHz.

For me this rather looks like a sign of an instable local oscillator somewhere in the three down conversion processes. There seems to be a lot of discrete 'levels' of the offsets, which might point to an oscillator that switches between discrete frequencies which then might have multiplied by a certain factor. I would probably try to produce a histogram of all the offsets and check whether this observation can be correct. If my suspicion is correct this would have consequences for adding up spectra at a certain zenith angle over a longer period as is implemented in the measurement scheme.

Anyway, I would like the authors to elaborate more on the large spread of the frequency offsets.

[in RM and TCF: Section 4.1. Spectrum setup and 5. Conclusions]

Based on feedback from you and Referee #2, we carefully reexamined this issue.

We identified that the offset is not caused by the Doppler effect but instead originates from an incorrect channel definition in the AC240 spectrometer and a frequency offset in the local oscillator.

According to Benz et al. (2005), the AC240 spectrometer analyzes a 1 GHz bandwidth using 16384 channels. As stated in Eq. (2) of the reference, the center frequency ($\nu$) corresponding to a channel is calculated as:

$$\nu_{AC240} = \frac{m}{16384} + \frac{0.5}{16384} \quad (m = 0, 1, 2, \dots, 16383)$$

where $m$ represents the channels for positive frequencies and ranges from 0 to 16383. As the frequency in Eq. (2) was defined as the lower edge of channel $m$, 0.5/16384 was added to represent the center frequency of the channel. However, in this study, the first channel of the spectrometer was incorrectly defined as 1 instead of 0, and the frequency was calculated by dividing the channel number, $k$, by 16384. Therefore, the corrected frequency at the spectrometer of SORAS should be calculated as follows:

$$\nu_{soras} = \frac{k-1}{16384} + \frac{0.5}{16384} = \frac{k-0.5}{16384} \quad (k = 1, 2, 3, \dots, 16384)$$

The corrected frequency is 30.5 kHz, smaller than the incorrectly defined frequency, and Fig. 9 and Fig. 10 have been revised to reflect the new frequency.

This adjustment further increased the offset from −55.1 kHz to −85.6 kHz.

Following your suggestion, we also plotted a histogram of the offsets. The histogram shows symmetry around the mean offset of −86 kHz. However, there is a significant frequency of offsets at −245 kHz and 60 kHz. As you mentioned, this appears to result from frequency shifts to specific levels caused by instability in the local oscillator, although it is unclear at which stage of the frequency conversion this occurs. Additionally, as you pointed out, such LO frequency shifts are likely to have a significant impact when averaging long-term observation data.

[Figure]

Figure 3. (Left) The individual frequency offset is determined by the curve fitting method. On average, the center frequency of the measurement is shifted by −85.6 kHz from 110.836 GHz. (Right) The frequency offset distribution is presented as a histogram, showing high occurrences of frequency shifts at −245 kHz and 60 kHz.

We examined the local oscillator of the baseband converter. The LO frequency has a frequency offset of −6 kHz from 2 GHz. The −6 kHz LO offset contributes to a −85.6 kHz shift.

We have incorporated it into the manuscript.

The content in the manuscript referring to the Doppler effect has been removed, and the text was revised to prevent further misunderstanding. Unlike previous studies investigating the Doppler effect with radiometers, such as Rufenacht et al. (2012) (6.1 kHz) and Hagen et al. (2018) (12.2 kHz), the frequency resolution in this study is 5 to 10 times larger. Additionally, due to the lack of opposite-direction observation corrections, the observation conditions in this study are insufficient to evaluate the Doppler effect.

[Reference]

Benz, A. O., Grigis, P. C., Hungerbühler, V., Meyer, H., Monstein, C., Stuber, B., and Zardet, D.: A broadband FFT spectrometer for radio and millimeter astronomy, Astron. Astrophys., 442, 767–773, https://doi.org/10.1051/0004-6361:20053568, 2005.

Rüfenacht, R., Kampfer, N., and Murk, A.: First middle-atmospheric zonal wind profile measurements with a new ground-based microwave Doppler-spectro-radiometer, Atmos. Meas. Tech., 5, 2647–2659, https://doi.org/10.5194/amt-5-2647-2012, 2012.

Hagen, J., Murk, A., Rüfenacht, R., Khaykin, S., Hauchecorne, A., and Kämpfer, N.: WIRA-C: a compact 142-GHz-radiometer for continuous middle-atmospheric wind measurements, Atmos. Meas. Tech., 11, 5007–5024, https://doi.org/10.5194/amt-11-5007-2018, 2018.